# The Proof-of-Concept: The Transformation of Naphthalene and Its Derivatives into Decalin and Its Derivatives during Thermochemical Processing of Sewage Sludge

**Jacek Łyczko [1], Jacek A. Koziel [2], Chumki Banik [2] and Andrzej Białowiec [2,3,\*]**

[1] Department of Chemistry, Wrocław University of Environmental and Life Sciences, 50-375 Wrocław, Poland; jacek.lyczko@upwr.edu.pl

[2] Department of Agricultural and Biosystems Engineering, Iowa State University, Ames, IA 50011, USA; koziel@iastate.edu (J.A.K.); cbanik@iastate.edu (C.B.)

[3] Department of Applied Bioeconomy, Wrocław University of Environmental and Life Sciences, 50-375 Wrocław, Poland

\* Correspondence: andrzej.bialowiec@upwr.edu.pl; Tel.: +48-71-320-5973

**Abstract:** One solution for sewage sludge (SS) management is thermochemical treatment due to torrefaction and pyrolysis with biochar production. SS biochar may contain toxic volatile organic compounds (VOCs) and polyaromatic hydrocarbons (PAHs). This study aimed to determine the process temperature's influence on the qualitative PAHs emission from SS-biochar and the transformation of PAHs contained in SS. SS was torrefied/pyrolyzed under temperatures 200–600°C with 1 h residence time. The headspace solid-phase microextraction (SPME) combined with gas chromatography and mass spectrometry (HS-SPME-GC-MS) analytical procedure of VOCs and PAHs emission was applied. The highest abundance of numerous VOCs was found for torrefaction ranges of temperature. The increase of temperatures to the pyrolytic range decreased the presence of VOCs and PAHs in biochar. The most common VOCs emitted from thermally processed SS were acetone, 2-methylfuran, 2-butanone, 3-metylbutanal, benzene, decalin, and acetic acid. The naphthalene present in SS converted to decalin (and other decalin derivatives), which may lead to SS biochar being considered hazardous material.

**Keywords:** biochar; PAHs; municipal waste; pyrolysis; torrefaction; VOCs; waste management

## 1. Introduction

Sewage sludge (SS), being the main solid waste of the wastewater treatment process (Dai et al., 2014), is produced in large quantities on a global scale. The annual SS production in Europe reaches around 10.13 million tons [1], while in the US, 12.56 million tons [2]. In China, the generation of SS exceeds 25 million tons per year [3]. There are three main categories of methods of SS disposal: agricultural use, landfilling, and incineration [4]. However, new approaches, technologies, and techniques are under investigation and development, such as SS gasification, pyrolysis, torrefaction, hydrothermal carbonization, and hydrothermal liquefaction [5]. One promising option is the waste-to-carbon concept [6,7], where organic waste is converted to carbon materials: carbonized solid fuels [8], different types of biochar [9], and hydrochar [10]. The main goals of using thermochemical methods are the generation of carbonized solid fuels, the mitigation of contaminants in the environment, and the increase of nutrient content in biochar. Torrefaction [6] and pyrolysis [11] are the most used for biochar production from SS [12].

Researchers have widely analyzed the applicability of carbon materials in terms of potential implementation directions. Research on carbon materials, including biochar applications, has shown that it can be used with benefits, among others as a soil amendment, during environmental remediation, solid fuel, energy storage, composite production [13], and in other areas (environment detoxification [13,14], metals removal [15], nitrates removal [13], water desalination [16], and for biogas production enhancement [10].

Here, for the first time, we propose the thermochemical treatment of SS as a mitigation method for persistent organic pollutants (POPs), including polycyclic aromatic hydrocarbons (PAHs). It is known that SS contains numerous toxic organic compounds (e.g., phthalate esters, PAEs; polycyclic aromatic hydrocarbons, PAHs; bisphenol analogs; synthetic musks; alkylphenol polyethoxylates; ultraviolet stabilizers; pharmaceuticals; antibiotics; hormones; polybrominated diphenyl ethers, PBDE; organochlorine pesticides; perfluorinated compounds; polychlorinated biphenyls, PCB; and heavy metals [17–21].

However, a review of state of the art in the field of biochar revealed that due to thermal organic matter transformation, numerous volatile organic compounds (VOCs) are formed under various torrefaction and pyrolysis conditions [22–25]. Thus, biochar itself may be a source of pollutants. This is mostly because, during pyrolysis and torrefaction, some compounds may condense onto the porous biochar structure [26]. Some studies [25–30] report a wide variety of VOCs present in biochar.

Our recent study identified 84 different VOCs emitted from carbonized solid fuel obtained from raw refuse-derived fuel (RDF) [27]. The identified compounds have been organized into five groups: alkyl derivatives of two-ring aromatic hydrocarbon; alkyl derivatives of benzene or phenols; compounds that are generally considered as lower risk (e.g., naturally present in food); derivatives of heterocyclic amines; belonging to other groups, in some cases with unknown structure. Spokas et al. [25] reported 140 different compounds. The top five most frequently observed compounds were acetone, benzene, methylethylketone, toluene, and methyl acetate. The transformation of VOCs as a function of feedstock and process and resulting potential toxins is not well understood.

The organic matter decomposition and biochar formation occur in three stages depending on the temperature. The processes in the first stage of biomass torrefaction (< ~320 °C) are related to water evaporation, bond breakage, and the formation of carbonyl and carboxyl groups. The second stage (pyrolysis ~350 °C–~500 °C) entails the depolymerization, fragmentation, and secondary reactions, which pose the maximum mass loss of the feedstock. The final stage of pyrolysis (~500 °C–~700 °C) is responsible for the slow decomposition of biochar solid residues with the parallel occurrence of the condensation of the polycyclic structures. The temperature increase leads to expanding the size and the degree of condensations of polycyclic groups [31]. The temperature is the most crucial factor in the generation of VOCs in biochars. Organic matter degradation starts at temperatures of just above 120 °C, subjected to char and gas formation at temperatures below 300 °C. The carboxyl and carbonyl groups are generated and subsequently decomposed $CO_2$ and CO during the charring process. The formation of hydroquinone, catechol, and phenol occurs from the partial degradation of biochar components such as lignin and cellulose. Moreover, PAHs are formed during the biochar production process [31]. PAHs production occurs in the processes of dealkylation, dehydrogenation, cyclization, aromatization, and/or radical reactions [32].

A similar trend was confirmed by Spokas et al. [25], who researched biochar originating from various substrates produced at various process temperatures. They showed that below 350 °C, produced biochars consisted of short carbon chain aldehydes, furans, and ketones. The temperatures above 350 °C produced biochars that contained mostly longer carbon chain hydrocarbons and aromatic compounds. A relationship was observed that the increase of the process temperature leads to decreased VOCs emission from biochars. Wang et al. [33], analyzing the PAHs content in biochar, arrived at similar conclusions. The lowest concentration of PAHs was found under conditions of slow pyrolysis and with longer retention (inside reactor) time. Ghidotti et al. [26] tested seven biochars

produced due to pyrolysis of corn stalk at increasing pyrolysis temperatures (350–650 °C). In total, 88 compounds were tentatively identified. The decreasing trend emerged between the number of compound classes and the decreasing H/C ratio and volatile matter content. Biochars with H/C <0.70 (>320 °C) did not release VOCs at ambient temperatures (25 °C).

We hypothesize that the increase of the thermochemical process temperature will mitigate the PAHs emission from biochar produced from SS and that the increase of the process temperature will influence the PAHs (present in raw SS) transformation into less harmful substances.

This study aimed to determine (1) the thermochemical process temperature's influence on the qualitative PAHs emission from biochar produced from sewage sludge and (2) the transformation of PAHs from sewage sludge to less harmful compounds. In this way, we intend to address the central scientific question of the PAHs' presence and formation that may be controlled by modification of the thermochemical (torrefaction and pyrolysis) process temperature.

## 2. Materials and Methods

### 2.1. Sewage Sludge Used for Experiments

SS for experiments was acquired from Jimmy Smith Wastewater Treatment Facility (WTF), Boone, IA, USA. This facility mechanically and biologically treats wastewater. The WTF capacity is 18.25 million $dm^3 \cdot d^{-1}$, but it treats ~9.4 million $dm^3 \cdot d^{-1}$. The first stage of mechanical treatment is screening through two automatic bar screens. The next stage is grit and grease separation. After that, the biological treatment starts in two oxidative ditches, where organic matter is removed and ammonia nitrogen is nitrified. The effluent is directed into two circular clarifiers for activated sludge sedimentation, and then the supernatant goes to the four tertiary sand filters for final solids removal. The last treatment stage is disinfection with ultraviolet light. Purified sewage is discharged to the South Fork New River. Produced SS is dewatered by a 2 m belt press and dried. The dewatered SS with 87–90% solids content is distributed to the public for use as fertilizer [34]. SS organic matter content, determined according to PN-EN 15169: 201126 and expressed as losses on the ignition, was 62.9% of dry matter (Table 1).

**Table 1.** The losses on ignition of raw SS and biochars samples produced from SS under different temperatures.

| Sewage Sludge Sample Type | Losses on Ignition, % d.m. |
| --- | --- |
| Raw sewage sludge | 62.9 |
| SS biochar under 200 °C | 61.0 |
| SS biochar under 220 °C | 60.5 |
| SS biochar under 240 °C | 59.2 |
| SS biochar under 260 °C | 57.5 |
| SS biochar under 280 °C | 54.7 |
| SS biochar under 300 °C | 60.9 |
| SS biochar under 450 °C | 33.7 |
| SS biochar under 600 °C | 31.9 |

### 2.2. Biochar Production Procedure

The SS sample was processed in a 150 mL steel crucible (Figure 1).

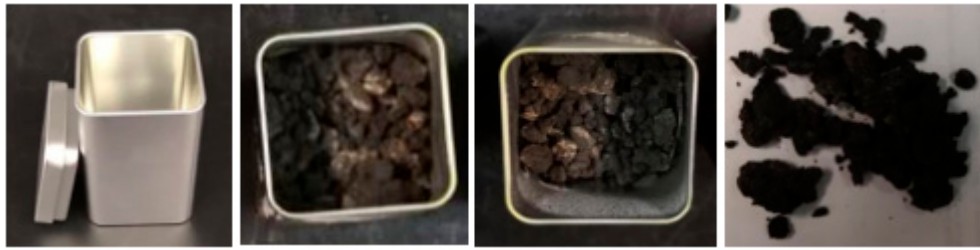

**Figure 1.** The sewage sludge and biochar samples example. From the left, the crucible, raw SS sample in the crucible, biochar from SS in the crucible, and produced biochar from SS.

The biochar was obtained using a muffle furnace. All samples were first dried at 105 °C. SS samples (223 g mean weight) were then placed in the reactor and heated for up to 1 h at constant temperatures of 200, 220, 240, 260, 280, 300, 450, and 600 °C, respectively. A technical gas, nitrogen, was connected to the apparatus to ensure inert conditions of the process. The gas flow set point was 1,000 mL.min$^{-1}$ during the first 5 min of the process and 40 mL·min$^{-1}$ after that. The heating process commenced 5 min after the gas was introduced into the apparatus. The gas cut-off occurred when the temperature inside the reactor was 100 °C. The mass yields of the biochar are given in Figure 2.

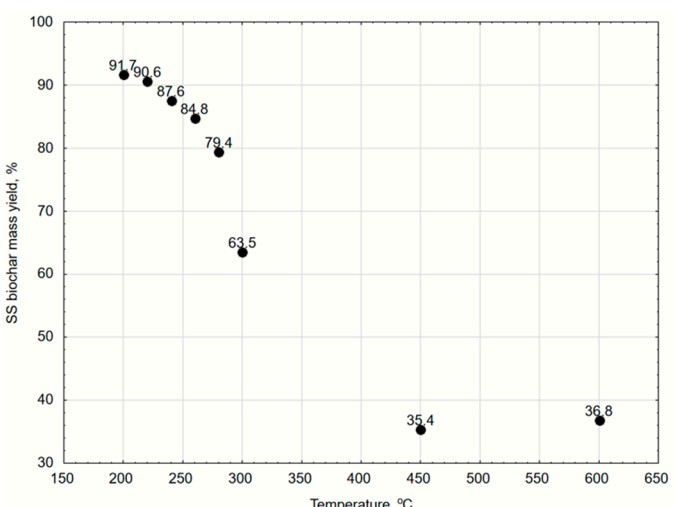

**Figure 2.** The mass yield of the biochar produced under different torrefaction/pyrolysis temperatures.

### 2.3. Solid-Phase Microextraction

The whole experimental procedure has been given in Figure 3. VOCs emitted from biochar were collected from headspace (HS) using solid-phase microextraction (SPME). Prior to SPME, ~1 g of the biochar or raw SS samples was transferred to a 20 mL amber glass vial (Microliter, Wheaton, Millville, NJ, USA) with a spatula. The vial was closed with an airtight half-hole with a PTFE-lined septum. SPME fiber (PDMS/DVB/Carboxen, 1 cm long, 50/30 μm film thickness) was then introduced into the HS, and VOCs were extracted for 30 min. The SPME fiber was then introduced into a heated GC injector, followed by separation with gas chromatography combined with mass spectrometry (GC–MS).

### 2.4. Chromatographic Analysis of Adsorbed VOCs by GC-MS

Custom multidimensional GC-MS (Microanalytics, Round Rock, TX, USA) was used for analyses of the raw SS and biochar samples. The GC-MS is based on the Agilent 6890 (GC) and 5973 (MS) (Agilent Technologies, Santa Clara, CA, USA). The chromatographic part had two capillary columns: a non-polar column (internal diameter 30 m × 0.53 mm ×

0.5 µm) with a fixed limiting column and a second polar column (internal diameter 30 m × 0.53 mm × 0.5 µm thickness) connected in series. System was controlled by automation and data collection software (Multitrax v.6.00.1, Microanalytics, Round Rock, TX, USA) and ChemStation E.01.01.335 (Agilent Technologies, Santa Clara, CA, USA). The run parameters were GC splitless inlet at 260 °C, 40 °C initial oven temperature was held for 3.0 min, then 7 °C ·min$^{-1}$ ramping to 240 °C, where it was held for 8.43 min. The total runtime was 40 min. Ultra-high purity He (99.999%, Airgas, Des Moines, IA, USA) was used as the carrier gas. Full scan range mass to charge ratio (*m/z*) was set from 34 to 350. Electron ionization (EI) mode was set to ionization anodes at 70 eV after scan acquisition.

Compound identification was completed by aligning the mass spectra of unknown compounds with the MS BenchTop/PBM library (Palisade Mass Spectrometry, Ithaca, NY, USA) and the NIST 17 Mass Spectral and Retention Index Libraries (NIST17) (The National Institute of Standards and Technology, Gaithersburg, MD, USA), and the Automated Mass Spectral Deconvolution and Identification System (AMDIS) (NIST, Gaithersburg, MD, USA). The semi-quantification, presented as a percentage of particular compounds, was based on compounds' relative abundances. All chromatograms have been provided as Supplementary Materials, file Supplementary Materials.pdf.

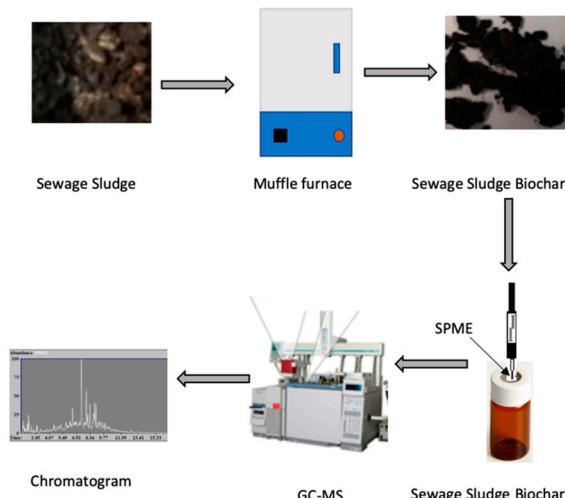

**Figure 3.** The experimental procedure.

## 3. Results and Discussion

### 3.1. Torrefaction and Pyrolysis of Sewage Sludge

The executed torrefaction and pyrolysis procedure resulted in a typical [35] decrease in the organic matter (OM) content in biochar samples with the temperature increase (Table 1). The losses on ignition of raw SS samples were 62.3% d.m. The biochars samples produced under torrefaction conditions (200–300 °C) contained organic matter in the range of 54.7–61.0% d.m. However, the temperature increase to 450 °C and 600 °C reduced the OM content by about 50% (Table 1).

The mass yield decreased gradually from 91.7% to 63.5%, while the temperature increased from 200 °C to 300 °C. The increase of temperature to the typical pyrolytic range caused the mass yield of biochar to be only ~36% of the initial mass of SS (Figure 2). The decrease of mass yield with the temperature affected the OM decomposition and volatilization of gaseous products, which was confirmed by a high correlation coefficient value of 0.913 between values of mass yield and losses on ignition.

### 3.2. VOC Emission for SS and Biochars

Sixty VOCs were found by HS-SPME-GC/MS analysis of eight biochar samples obtained in different temperature conditions and sewage sludge used for their production.

Among them, 48 were successfully identified, and 10 were unidentified. However, they were qualified to either the naphthalene or decalin derivatives group, and two were unidentified. Identified compounds may be qualified to various chemical groups such as aldehydes (e.g., 2-methylbutanal, hexanal), ketones (e.g., acetone, 2-butanone–KME), PAHs (e.g., naphthalene derivatives), and hydrogenated PAHs derivatives (e.g., decalin, 1-methyldecalin, trans-2-methyldecaline).

VOCs that occurred most frequently in the samples were acetone (found in seven of nine samples), 2-methylfuran (found in five of nine samples), 2-butanone (found in five of nine samples), 3-methylbutanal (found in five of nine samples), benzene (found in six of nine samples), decalin (found in five of nine samples), and acetic acid (found in six of nine samples); Table 2. In terms of contribution in biochar sample vapors, the highest was 2-amino-5-methylbenzoic acid, which was the only VOC found and identified in SS450 and SS600 biochar samples. Further, the highest contribution was observed for unidentified trans-2-methyldecalin (26.13%), limonene (23.58%), and decalin (23.44%) in SS300, SS Raw, and SS200 biochar samples, respectively. Regarding the number of unique compounds, the largest observed group was decalin derivatives, consisting of sixteen VOCs.

**Table 2.** The changes in volatile organic compound profiles caused by different temperatures during the biochar production process.

| Compound | LRI [1] | CAS | SS RAW | SS200 | SS220 | SS240 | SS260 | SS280 | SS300 | SS450 | SS600 |
|---|---|---|---|---|---|---|---|---|---|---|---|
| | | | Contribution (%) | | | | | | | | |
| (*E*)-2-Butene | 450 | 0107-01-07 | - | - | - | 4.96 | - | - | - | - | - |
| *n*-Hexane | 600 | 110-54-3 | - | - | - | - | 0.68 | - | - | - | - |
| Acetaldehyde | 702 | 75-07-0 | - | - | 1.17 | - | - | - | - | - | - |
| (*Z*)-4-Methyl-2-hexene | Na [2] | 3683-19-0 | - | - | - | - | 1.08 | - | - | - | - |
| Carbon disulfide | 735 | 75-15-0 | - | - | 1.89 | - | - | - | - | - | - |
| 2,4,4-Trimethyl-1-pentene | 750 | 107-39-1 | - | - | - | 7.31 | 11.84 | 7.69 | - | - | - |
| 2,4,4-Trimethyl-2-pentene, | 755 | 107-40-4 | - | - | - | - | - | 1.98 | - | - | - |
| (*E*)-4-methyl- -2-hepten | 797 | 66225-17-0 | - | - | 2.97 | 7.74 | 19.20 | - | - | - | - |
| Furan | 799 | 110-00-9 | - | 2.94 | 7.17 | 4.92 | - | 0.68 | - | - | - |
| (*Z*)-1,3-Dimethylcyclohexane | 810 | 638-04-0 | - | - | - | 1.47 | 2.20 | - | - | - | - |
| Acetone | 819 | 67-64-1 | 10.89 | 6.53 | 7.36 | 4.43 | 2.80 | 3.19 | 4.82 | | |
| Acetic acid, methyl ester | 828 | 79-20-9 | - | 3.08 | 3.72 | 12.48 | - | 71.54 | - | - | - |
| (E)-4-Octene | 840 | 14850-23-8 | - | - | - | - | 0.32 | - | - | - | - |
| 2-Methyl-furan | 869 | 534-22-5 | - | 3.76 | 5.55 | 9.95 | 8.53 | 2.09 | - | - | - |
| 2-Butanone | 907 | 78-93-3 | - | 1.90 | 1.93 | 2.11 | 2.08 | 2.04 | - | - | - |
| 2-Methylbutanal | 914 | 96-17-3 | - | 0.42 | 0.60 | | | 0.37 | - | - | - |
| 3-Methylbutanal | 918 | 590-86-3 | - | 0.34 | 0.65 | 0.66 | 1.15 | 0.03 | - | - | - |
| 3-Methyl-2-butanone | 930 | 563-80-4 | - | 0.17 | - | - | - | - | - | - | - |
| Ethanol | 932 | 64-17-5 | - | 3.12 | 0.25 | - | - | 2.68 | - | - | - |
| Benzene | 957 | 71-43-2 | 2.53 | 4.24 | 7.97 | 4.05 | 2.07 | 0.68 | - | - | - |
| Methyl isobutyl ketone | 1010 | 108-10-1 | - | - | 0.92 | - | - | - | - | - | - |

| Compound | RI | CAS | | | | | | | | | |
|---|---|---|---|---|---|---|---|---|---|---|---|
| Acetonitrile | 1013 | 75-05-8 | - | 0.83 | - | - | - | - | - | - | - |
| Glycolaldehyde dimethyl acetal | na | 621-63-6 | 17.13 | - | - | 1.09 | 0.47 | - | - | - | - |
| Trimethylsilanol | na | 1066-40-6 | - | - | - | - | - | - | 7.24 | - | - |
| Toluene | 1042 | 108-88-3 | - | 0.14 | 0.46 | - | - | - | - | - | - |
| unknown | | - | - | 2.63 | 2.25 | - | - | - | - | - | - |
| (*Z*)-1-Ethyl-1,3-dimethyl-cyclohexane | na | na | - | 0.35 | 2.07 | 0.79 | 0.18 | - | - | - | - |
| Dimethyl disulfide | 1077 | 624-92-0 | - | 1.43 | - | - | 0.94 | - | - | - | - |
| Hexanal | 1083 | 66-25-1 | - | - | 1.06 | - | - | - | 1.27 | - | - |
| 2-Propen-1-ol | 1123 | 107-18-6 | - | - | 0.05 | - | - | - | - | - | - |
| Butanenitrile | 1129 | 109-74-0 | - | - | 0.50 | - | - | - | - | - | - |
| 1-(2-Furanyl-)2-pentanone | na | 20907-03-3 | - | - | - | 0.09 | - | - | - | - | - |
| Naphtalene derivative | 1170 | - | - | - | 8.91 | - | - | - | 4.66 | - | - |
| Decalin | 1170 | 91-17-8 | - | 23.44 | 12.21 | 3.49 | 1.99 | 1.03 | | - | - |
| (*E*)-2-Methyl-decalin | na | 2958-76-1 | 15.89 | - | 0.57 | 10.43 | 8.44 | | 26.13 | - | - |
| Limonene | 1200 | 5989-54-8 | 23.58 | - | - | - | - | - | - | - | - |
| Naphtalene derivative | | - | - | - | 0.53 | - | - | - | - | - | - |
| 1-Methyldecalin | 1215 | 2958-75-0 | - | 11.27 | 0.15 | - | - | 0.70 | - | - | - |
| Decalin derivative | - | - | - | 4.39 | 6.60 | - | - | 0.28 | 11.54 | - | - |
| Phthalic acid, cyclobutyl heptyl ester | na | na | 0.21 | - | - | - | - | - | - | - | - |
| (*E*)-4a-Methyldecalin | na | 2547-27-5 | 8.97 | - | - | - | - | - | - | - | - |
| (*E*)-9-Methyldecalin | na | na | - | - | - | 5.65 | 5.50 | - | 2.82 | - | - |
| 2,6-Dimethyldecalin | 1226 | 1618-22-0 | - | 6.80 | 4.31 | 2.68 | | 0.32 | | - | - |
| Decalin derivative | - | - | - | - | - | - | 1.78 | - | 5.58 | - | - |
| 1,5-Dimethyldecalin | na | 66552-62-3 | 3.87 | - | - | 3.48 | 1.48 | - | - | - | - |
| Decalin dimethyl-derivative | - | - | - | 2.57 | 3.41 | - | - | 0.12 | - | - | - |
| Decalin dimethyl-derivative | - | - | - | 5.71 | 7.47 | - | - | 0.45 | 6.51 | - | - |
| Decalin dimethyl-derivative | - | - | - | 7.46 | 1.25 | - | 0.69 | 0.55 | 6.09 | - | - |
| 1,6-Dimethyldecalin | na | 1618-22-0 | - | - | - | - | 0.83 | - | 12.52 | - | - |
| 2,3-Dimethyldecalin | na | na | 5.81 | - | - | 2.27 | 1.73 | - | - | - | - |
| Decalin derivative | - | - | 3.32 | 1.21 | 0.94 | - | - | - | - | - | - |
| Decalin dimethyl-derivative | - | - | - | 1.00 | 2.82 | 3.05 | 4.17 | - | - | - | - |
| unknown | - | - | - | 0.41 | 0.03 | - | - | - | - | - | - |
| (*Z,E*)-3-Ethylbicyclo[4.4.0]decane | 1301 | na | - | 3.42 | - | - | - | - | - | - | - |
| Naphtalene derivative | - | - | - | - | 0.05 | - | - | - | - | - | - |
| Acetic acid | 1449 | 64-19-7 | 1.82 | 0.45 | 2.24 | 6.37 | 17.55 | 3.59 | 10.84 | - | - |
| Furfural | 1461 | 98-01-1 | - | - | - | 0.53 | 2.30 | - | - | - | - |
| Methoxyphenyloxime | na | na | 4.15 | - | - | - | - | - | - | - | - |

| | | | | | | | | | | | |
|---|---|---|---|---|---|---|---|---|---|---|---|
| 2-Amino-5-methylbenzoic acid * | na | 2941-78-8 | - | - | - | - | - | - | - | 100 | 100 |
| 2-Amino-3-chloro-1,4-naphthalenedione | na | 2797-51-5 | 1.82 | - | - | - | - | - | - | - | - |

[1] LRI-linear retention indices according to NIST 17 Mass Spectral and Retention Index Libraries (NIST17); [2] na—not available; * tentatively identified by a mass spectrum match of 60%.

Most of the VOCs found in biochar vapors present a significant hazard potential, both in human health and environmental risks. Especially worth considering were high contributions of naphthalene derivatives (4.66% in SS300 biochar and 9.49% in SS220 biochar total vapors), decalin (from 1,03% up to 23.44% of biochar samples vapors), and decalin derivatives (from 0.12% up to 26.13% biochar samples vapors). According to Stuchal et al.'s [36] research, decalin may cause hepatotoxicity when continuously inhaled at a level of 7.9 mg·m$^{-3}$, while Clewell et al. [37] report that 1.58 mg·m$^{-3}$ continuous exposure of naphthalene vapors may cause changes in the nasal epithelium tissue. Additionally, early research regarding decalin toxicity in terms of nephrotoxicity was performed by Stone et al. [38]. Moreover, according to the Centers for Disease Control and Prevention (CDC) database, naphthalene and decalin present extremely low lower explosive limits (LEL) specified as 0.9% and 0.7%, respectively.

The issue regarding VOCs emissions from various biochar products being potentially dangerous for life, health, and the environment was considered in our earlier research [27,39,40] focused on carbonized refuse-derived fuel (CRDF) obtained from municipal solid waste (MSW). In most cases, studies focus on adsorbing VOCs with biochar application [33,41,42]. The importance of VOC emissions from biochar is underestimated. Therefore, the possibilities of reducing VOCs emission from biochar during storage or transport should be more accurately investigated for the protection of human and environmental wellbeing. Regarding the profile of VOCs distribution (Table 2), the contribution of VOCs in biochar sample vapors was strongly dependent on the production process conditions. The trend of changes was perfectly pictured in samples SS200, SS220, SS240, SS260, SS280, SS300, SS450, and SS600 by changes in decalin contribution—23.44%, 12.21%, 3.49%, 1.99%, 1.03%, 0.0%, and 0.0%, respectively. Such results correlate with findings of Chen et al. [17], where it was found that increasing temperature during the biochar production process reduces the PAHs contribution and supports the conclusion that the pyrolysis, rather than torrefaction, may be considered as a tool for reducing hazardous VOCs emission from SS-based biochar.

Higher contribution of decalin and its derivatives than naphthalene and other PAHs in biochar samples vapors were found as a surprising result since, in most studies, PAHs are considered as the main SS-based biochar risk factor [43,44]. Previous research considered the detection of PAHs in liquid extracts of SS-derived biochar. In this study, the headspace of SS-derived biochar samples was, to our best knowledge for the first time, examined by HS-SPME-GC-MS, which was established as a useful tool for evaluating VOCs emission from biochar [27]. Considering physical properties–boiling point, decalin (190 °C) is much more volatile than naphthalene (218 °C), which shows that the HS-SPME technique is more specific for this compound. Nevertheless, it is important to assess if PAHs are the most urgent risk regarding SS-based biochar.

### 3.3. PAHs Transformation

SS is a material that may be characterized by metal content, including Ni [45,46], which is used as a catalyst for the hydrogenation process [47]. As described by Feiner et al. [47] in a model experiment, with temperatures adequate for the torrefaction process, the presence of catalysts (such as Ni) and elevated pressure lead to the conversion of naphthalene to decalin (via an intermediate step—tetraline). Comparing this result may explain that our non-targeted (for PAHs) chemical analysis revealed that decalin and its deriva-

tives are the main representatives in SS-based biochar profiles. Nevertheless, some concerns regarding transforming PAHs to decalin during the temperature processes should be considered. In this research, contrary to Feiner et al. [47], no apparent source of hydrogen may be pointed out, and the process of biochar production was performed under atmospheric pressure. However, a hypothesis supporting the idea of transformation of PAHs to decalin may be proposed. Additionally, from a practical point of view, the pretreatment of SS by heavy metals leaching, including Ni, before pyrolysis, may decrease decalin formation if Ni catalases decalin formation.

Regarding hydrogen source, some organic constituents of SS may be decomposed to molecular hydrogen with increasing process temperature, which may explain the possibility of saturating bonds in PAHs structure (Figure 4). In turn, the pressure issue may be explained by clustering the material during the thermal process. Formed in this way, sinters may cause local pressure elevation inside of them, which improves the efficiency of PAHs transformation. Nevertheless, this must be verified. The temperature increase during SS-based biochar still efficiently reduces transformation into PAHs and the contribution of decalin and its derivatives.

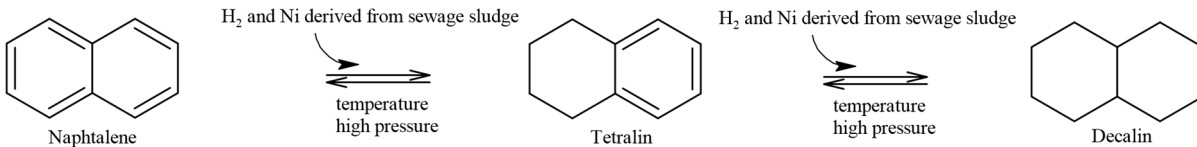

**Figure 4.** Transformation process of naphtalene into decalin.

## 4. Conclusions

The experiment revealed for the first time that pyrolysis might be used for the mitigation of volatile organic compounds and polyaromatic hydrocarbons emission from biochars produced from sewage sludge. It has also been revealed that sewage sludge torrefaction is not an effective method for mitigating volatile organic compounds and polyaromatic hydrocarbons emissions from biochar. The lower temperature increased the number of present volatile organic compounds and polyaromatic hydrocarbons (including derivatives). The most frequently found compounds in thermally processed sewage sludge were acetone, 2-methylfuran, 2-butanone, 3-methylbutanal, benzene, decalin, and acetic acid. Additionally, for the first time, we have found that the naphthalene was converted to decalin and other decalin derivatives, more toxic and volatile compounds, during torrefaction. The identified risk of higher contamination and toxicity of sewage-sludge-based biochar compared to raw sewage sludge requires further investigation on the development and application of sewage sludge torrefaction technologies. Additional research is required for a more comprehensive assessment of sewage-sludge-based biochar produced via torrefaction, including its potential impact on the environment.

**Supplementary Materials:** The following are available online at www.mdpi.com/article/10.3390/en14206479/s1, file Supplementary materials.pdf.

**Author Contributions:** Conceptualization, A.B.; methodology, J.Ł., J.A.K., and A.B.; validation, J.A.K. and A.B..; formal analysis, J.Ł.; investigation, C.B. and A.B.; resources, J.A.K.; data curation, J.Ł. and A.B.; writing—original draft preparation, J.Ł. and A.B.; writing—review and editing, J.Ł. and A.B.; visualization, J.Ł. and A.B.; supervision, J.A.K. and A.B.; project administration, A.B.; funding acquisition, A.B. All authors have read and agreed to the published version of the manuscript.

**Funding:** The authors would like to thank the Fulbright Foundation for funding the project titled "Research on pollutants emission from Carbonized Refuse Derived Fuel into the environment," completed at Iowa State University. In addition, this project was partially supported by the Iowa Agriculture and Home Economics Experiment Station, Ames, Iowa. Project no. IOW05400 (Animal Production Systems: Synthesis of Methods to Determine Triple Bottom Line Sustainability from

Findings of Reductionist Research) sponsored by Hatch Act and State of Iowa funds. The presented article results were obtained as part of the activity of the leading research team—Waste and Biomass Valorization Group (WBVG). The publication is financed under the Leading Research Groups support project from the subsidy increased for the period 2020–2025 in the amount of 2% of the subsidy referred to Art. 387 (3) of the Law of 20 July 2018 on Higher Education and Science, obtained in 2019.

**Data Availability Statement:** Data is contained within the article or supplementary materials.

**Conflicts of Interest:** The authors declare no conflicts of interest.

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
