# Peer review of "The Proof-of-Concept: The Transformation of Naphthalene and Its Derivatives into Decalin and Its Derivatives during Thermochemical Processing of Sewage Sludge"

_energies, doi:10.3390/en14206479_

Round 1

Reviewer 1 Report

The authors studied the process temperature influence on the qualitative PAHs emission from SS-biochar and on the transformation of PAHs contained in sewage sludge (SS). The increase of temperatures to the pyrolytic range decreased the presence of VOCs and PAHs in biochar. Main VOCs in the samples were acetone, 2-methylfuran, 2-butanone, 3-metylbutanal, benzene, decalin, and acetic acid. The authors also noted that the naphthalene present in SS converted to decalin and its derivatives which is even more toxic. Results are interesting potentially useful in the future for Sewage sludge management. I would recommend for publication after minor revisions

  1. The introduction is poorly organized. Line 33 doesn't provide any pieces of information on new technologies or techniques.
  2.  Authors found that naphthalene is converted to decalin. What are the harmful effects of decalin on the human body and environment?
  3. Fig. 3 suggests that H2 and Ni are needed for the above conversion. Is it possible to remove at least Ni before heating?

Reviewer 2 Report

Dear Sir,
The paper is informatie and well prepared however it can be improved in some areas as follows:

In the abstract

  • The full name of any abbreviation should be stated first such as HS-SPME-GC-MS
  • keywords should be arranged alphabetically
  • in other sections:

Some grammatical error should be corrected and there is some typos in the manuscript for example:

  • Line 40 The literature should be literature
  • Line 121 dm3 correct it to superscript
  • Schematic diagram for the experimental part could be useful for the reader
  • Did the author try different thermal parameters during the GC analysis as 40 minutes as working time considered long time.
  • Could the author explain his postulates about employing quanitative measurements.
  • I think that adding a chromatogram into the manuscript could be useful.
  • Please specify how many replicates did the authors perform for the analysis.
  • Reference should be checked carefully according to the style which is specified by the journal.

Regards
